# How Stable Are Individual Preferences for Health Expenditure in Germany?

**DOI:** 10.3390/healthcare13091074

**Published:** 2025-05-06

**Authors:** Bodo Vogt, Andreas Uphaus

**Affiliations:** 1Department of Business and Medical Faculty, Otto-von-Guericke-University Magdeburg, Universitätsplatz 2, 39106 Magdeburg, Germany; 2CareTech OWL, Bielefeld School of Business, University of Applied Sciences and Arts Bielefeld (HSBI), Interaktion 1, 33619 Bielefeld, Germany

**Keywords:** patient preferences, allocation, health system, COVID-19

## Abstract

Background/Objectives: This study analyzed the stability of individual preferences for the allocation of expenditure in the healthcare system using an experimental setting. Understanding these preferences can support policy decisions aimed at achieving a more needs-based allocation of scarce resources in healthcare systems. Stability in preferences might be essential in order to avoid frequent legislative changes and can potentially enhance public satisfaction with the healthcare system. Methods: Individual preferences were assessed through two questionnaire-based experimental studies conducted before and after the COVID-19 pandemic, each with about 160 participants, in the context of a healthcare seminar in the MaxLab of the Otto-von-Guericke-University Magdeburg, Germany. This study was intended as a preliminary study for a larger follow-up panel study. In particular, the questionnaire contained questions regarding satisfaction with the healthcare system, optimization options, possible maximum contributions, and preferences for the allocation of notional healthcare budget and research funds in order to provide initial evidence regarding the stability of such preferences. As the data were collected both before and after the COVID-19 pandemic, this significant change in the situation helps to provide clear indications of stability. The preferences collected were compared to the actual allocation of expenditure derived from official statistics in order to identify potential areas for policy adjustment. Results: Preferences for the allocation of healthcare expenditure appear to be relatively stable despite the effects of the pandemic. However, noticeable discrepancies exist between individual preferences and actual healthcare spending. Satisfaction with the healthcare system also remains relatively stable at a high level. Conclusions: Overall, the scientific measurement of public preferences could support more informed political decision-making and contribute to sustained satisfaction with the healthcare system. In particular, the distribution of funds to different disease categories should be adjusted on the basis of such preferences, taking into account the respective medical indications after representative regular surveys have been carried out.

## 1. Introduction

### 1.1. Key Issues

Healthcare expenditure accounts for a high proportion of GDP, having risen sharply (in absolute and per capita terms) in both the USA and Germany in recent years [1,2]. Given the scarcity of resources associated with rising expenditure, the partial consideration of patient preferences in the allocation of resources appears to be advantageous.

The German healthcare system is very complex, including both statutory health insurance (SHI) and private health insurance (PHI) [3,4]. While PHI is, in principle, a free market, German SHI is more of a regulated supply system. Approximately 88% of the working population is covered by SHI [5]. For this reason, in the following, we primarily look at the SHI framework, in which employees and their employers each currently pay 7.3% of their gross wages as contributions on a parity basis, supplemented in some cases by additional individual health insurance contributions. Therefore, health insurance contributions (and, thus, the total healthcare budget available for distribution) are limited.

Healthcare services are primarily provided by general practitioners (GPs) and specialists in private practices, hospitals, rehabilitation facilities, therapy centers, pharmacies, and nursing services. These services are billed, either directly or indirectly, via SHI and PHI funds. Approved services are defined by the Federal Joint Committee (FJC), which is an essential part of self-administration in the German healthcare system. However, the framework conditions are also set through policy via budgets.

Policymakers often allocate and prioritize resources on behalf of the population, either directly or indirectly [6]. The allocation of resources by elected decision-makers and experts is based on a variety of criteria; however, decisions are frequently made without sufficient knowledge of the general population’s preferences. While individual preferences often relate to specific treatment decisions, from a policymaking perspective, aggregated preferences of the general population appear to be relevant for the prioritization of treatment methods and, consequently, entire disease categories [7]. In the medical context, preferences are usually defined as those determining the choice between alternative treatments and services. Individuals should decide on alternatives based on their information, taking into account risks, benefits, and other aspects. Preferences can be measured in various ways [8]. In the context of this study, the alternatives primarily refer to different disease categories, while preferences are measured through the proportional direct allocation of monetary budgets.

Overall, the prices for healthcare services are, therefore, based on the supply-side distribution of expenditure and not-efficient equilibrium prices between supply by the state on one hand and the demand of patients on the other. Therefore, it seems advantageous to avoid serious differences with respect to demand preferences in the design of supply in order to optimize public satisfaction with the healthcare system and the efficiency of pricing. As a market solution does not appear to be immediately possible, it would, therefore, be beneficial not to have to constantly collect and integrate demand preferences; however, this is only possible if preferences are collected on a temporary basis and are fundamentally stable.

The first question that arises is to what extent the actual distribution of healthcare resources aligns with the preferences of individuals, that is, insured persons or patients. To address this question, the present study compared preferences derived from a questionnaire with the actual allocation of funds in practice. In addition to this comparison, this study examined how preferences and allocations may be influenced by significant external events.

The supply-oriented allocation system—in which the Federal Joint Committee (FJC, Gemeinsamer Bundesausschuss) of Germany regulates the framework conditions for the scope, quality, and pricing of services—is neither cost-effective nor based directly on market principles. Budgets are allocated to sectors such as hospitals, outpatient care, and pharmaceuticals [9] without directly considering patient preferences. However, understanding these preferences can help both policymakers and physicians to set appropriate priorities. Moreover, patient compliance with medical advice tends to increase when individuals are more satisfied with their care [10].

Previous studies have employed hypothetical budget allocations to assess public preferences, consistently finding substantial differences between those preferences and actual budget distributions. In such studies, citizens were placed in the role of political decision-makers [11]. The present study adopted a similar approach but focuses on preferences related to specific diseases rather than broader healthcare sectors.

### 1.2. Satisfaction

In addition to preference as a behavior-determining variable, satisfaction with the healthcare system is an important variable in behavioral economics that affects the functioning and quality of a healthcare system. One assumption that this study makes regarding patient satisfaction with a healthcare system is that it is related to the difference between actual allocation and patient preferences. This study also examines the influence of the COVID-19 pandemic as an influencing factor; that is, it considers whether the experiences of insured individuals during the COVID-19 pandemic have increased or decreased their satisfaction regarding the provision of healthcare.

In the following, patient preference is determined according to the total resources allocated to individual areas. This means that there is no direct dependency on supply, therefore making allocation and preference easier to compare. Further verbal supplementary information offered to the participants (which will be discussed later) was abstracted due to the individual nature of the cases and to avoid information overload in the survey. Consequently, the participants’ respective information bases are considered decisive for the determination of their allocation preferences.

We issued the study questionnaire twice, with an interval of three years between, which allowed us to observe developments in patient preferences. Notably, this period coincided with the COVID-19 pandemic. Thus, we further aimed to determine the extent to which their experiences during this period influenced their preferences.

### 1.3. COVID-19

The COVID-19 pandemic was an intense experience for many people, especially in terms of its effect on their approaches to health issues. The pandemic also intensified the focus of public debate on the dangers of viral diseases. From a research perspective, this event offers the opportunity to investigate how the actual allocation and preferences regarding healthcare services have changed. This makes it possible to examine a single determinant of patient preferences against actual allocation, allowing us to examine whether preferences and actual allocation are converging.

COVID-19 is a viral disease that emerged in December 2019, and the associated declaration of a health emergency was not lifted by the World Health Organization until March 2023. Worldwide, the number of deaths caused by COVID-19 has been estimated to be between 7 and 20 million [12]. The widespread of the disease severely overstretched the capacity of healthcare systems, leading to calls for greater system resilience [13], while supply chain problems led to resource shortages [14]. In instances of extreme resource shortages, ethics-based allocation principles (e.g., youngest or oldest first, health lotteries) were applied [14]. Another problem was that, due to an extreme reduction in employment relationships, employee-related health insurance was no longer in effect for many people [13].

### 1.4. Hypotheses

It is evident that the COVID-19 pandemic led to fundamental disruptions in healthcare systems worldwide. The question considered in this study relates to whether this effect has had an impact on resource allocation preferences. The attributes of preferences that lead to satisfaction include waiting and treatment times, information and shared decision-making, prices and the availability of the “best” treatments, the prevention and control of diseases, and safety [10,15,16,17]. For this reason, this study measures the influence of COVID-19 through a questionnaire, which was disseminated both before and toward the end of the pandemic.

The satisfaction of patients with their respective healthcare systems is also strongly influenced by the level of expenditure of the healthcare system, the density of the doctor network, and the ability to freely choose a doctor [14]. Public satisfaction with a healthcare system is also subject to change [9]. Such changes are often triggered by events [14], such as the COVID-19 pandemic.

While disease-specific preference studies do exist [14], a comprehensive analysis of the effect of diseases on preference formation has not yet been performed. Core hypotheses that should be rejected in this context have emerged regarding the significance of events such as the COVID-19 pandemic for the German population and their preferences, as follows (The statements under investigation are formulated as hypotheses. In the context of hypothesis testing, their opposites are specified as null hypotheses).

**Hypothesis 1 (H1).** 
*The COVID-19 pandemic has changed patient preferences in regard to the healthcare system, particularly as they relate to the prioritization of diseases. Approaches to viral infections are becoming more important.*


**(H1a).** 
*The preferences for budget allocation to address diseases will change, and the COVID-19 pandemic has had an influence on such allocation.*


**(H1b).** 
*The preferences for budget allocation to fund research into diseases will change, and the COVID-19 pandemic has had an influence on such allocation.*


**Hypothesis 2 (H2).** 
*Satisfaction with the healthcare system has suffered as a result of the COVID-19 pandemic, as the public perceives the state as caring for them.*


**(H2a).** 
*Satisfaction with the types of healthcare services has changed.*


**(H2b).** 
*The relative importance of satisfaction measures has changed.*


**Hypothesis 3 (H3).** 
*With appropriate regulation, it can be assumed that demand preferences and actual healthcare expenditure are in line, as the supply policy—which is created on behalf of the population—requires a high degree of acceptance to be enforceable.*


This study explored how individuals from a small German sample preferentially allocate healthcare spending across disease categories, as well as how the COVID-19 pandemic has affected these preferences, using an experimental approach. This approach allows insights into the determinants of such preferences to be obtained. In this context, the hypotheses presented above were tested. Finally, a comparison between the measured preferences and the actual allocation in Germany provides insights regarding changes that may increase the satisfaction of individuals with respect to healthcare spending allocation.

## 2. Materials and Methods

### 2.1. Study Design, Data Collection, and Procedures

The study was based on the demand-oriented allocation preferences of current and future health insurance and taxpayers. These individuals were surveyed as part of a questionnaire study. To address H3, their preferences are compared to real expenditure allocation data from official statistics. To address H1, data from before the pandemic were compared with corresponding data collected during and after the pandemic.

The questions on allocation preferences (F101–F110) are provided in Appendix A. To enable a more concrete comparison, the 12 most frequently occurring International Classification of Diseases (ICD) groups were selected, with all others being combined into a group named “other” (Appendix B). The ICD group designations were supplemented with explanatory examples, allowing them to be more easily classified by the respondents. The same questions were first asked before the COVID-19 pandemic (in April 2019) and then again after the COVID-19 pandemic (in April 2023) in the context of seminars given at Otto von Guericke University, Magdeburg. Students were tasked with conducting at least five surveys each. The COVID-19 pandemic occurred randomly between the planned study dates. The surveys were conducted online using Sawtooth Lighthouse Studio vers. 9.6.1 and 9.15.0.

The real medical costs in Germany were taken from the Federal Health Report 2024 [18]. These medical costs were then classified according to the ICD-10 categories and divided according to allocation preferences. Data on medical costs were collected infrequently; the most recent such data available were for the years 2015 and 2020 (with 2015 being four years before the start of the COVID-19 pandemic) [19]. The COVID-19 pandemic lasted from January 2020 to May 2023. The year 2020 was at the beginning of but completely within the pandemic period, meaning that any significant effects of the pandemic on costs were included in the available data. As the proportions have obviously remained similar, despite slight changes, it can be assumed that the proportions did not change significantly by the time of the respective surveys. An overall comparison of preferences and costs can, therefore, be made for the periods before, during, and after the pandemic, taking the above assumptions into account.

To calculate the costs per case, case numbers are required. The number of cases in 2021, by disease, was obtained from the German Federal Statistical Office (Destatis) [20] and Grube (Barmer) [21]. However, these numbers refer to cases in hospitals; that is, inpatient cases on the one hand and outpatient cases on the other hand separately due to significant differences.

### 2.2. Participants

The experimental studies were conducted in the Maxlab at the University of Magdeburg as part of a seminar with 33 (2019) and 34 (2023) students. Each student had the task of interviewing at least five acquaintances, who could be chosen freely. As a result of this design, other students and older adults were interviewed primarily at random. In this respect, the majority of random respondents belonged to the group of future contributors who have a particular interest in the development and allocation of health insurance expenditure. As we are interested in changes in expenditure in the future, the choice of younger people was considered to be important for this purpose. In total, 157 people took part in the 2019 survey and 163 in the 2023 one. The proportion of males was slightly above average. The participants mainly came from the wider area of the neighboring federal states. In terms of their level of education, most respondents held at least a Vocational Baccalaureate (2019: 73.8% and 2023: 78.0%). The participants’ characteristics are summarized in Table 1.

The number of participants in each group and the gender distribution were evidently comparable. The age of the respondents was expected to be relatively low, as most were students. This was intentional, as future payers of tax and health insurance premiums were the focus of this study.

Overall, this experimental preliminary study focused on students and their acquaintances and, thus, on a relatively young and well-educated sample. This was intended to include future contributors in particular but is not representative. Consequently, it was not possible (or intended) to draw conclusions about the entire population from the sample but merely to determine the basic effects in this target group. Through the choice of younger adults as participants, we expected to obtain better insights into expected future changes as the old die while the young demand more services. Nevertheless, 22.3% of respondents in the first study and 34.6% in the second were over 30 years old. A more representative study is planned for the future.

The participants in the 2019 group were somewhat younger than those in the 2023 group. This is consistent with the relatively lower household income in the 2019 group. The total number of hospitalizations appeared to be plausible (3–4). This number increased after the pandemic, which may be due to both the pandemic and the slightly higher age of the respondents. The number of visits to the doctor per year increased after the pandemic. Household income, doctor visits, and hospital stays were adjusted for outliers.

### 2.3. Measurement

The online questionnaire was essentially divided into 4 areas: willingness to pay, budget allocation, satisfaction, and personal questions (see also Appendix A). The answers were either specific amounts of money or percentages on a slider scale. The questionnaire was tested in advance with 4 different people in order to assess its comprehensibility and logic and was optimized accordingly.

In the questionnaire, the respondents’ willingness to pay (WTP) was measured to inform the composition of a potential budget to be allocated. WTP can be measured in different ways in this context [22]. The monetary evaluation, which focused on quantifying the benefit of the consideration, was limited. Preferences for the maximum insurance amount to be paid in EUR (F104) were determined. In addition, sociodemographic data regarding age, household income, gender, region, and school-leaving qualifications were collected.

To test H1 (“Preference changes due to COVID-19”), a fictitious healthcare budget of EUR 100 billion to address the viral disease (F109; H1a) and a fictitious additional research budget of EUR 100 million for disease research (F110; H1b) were determined. The sum of the amounts had to correspond to the budget, which was checked mathematically. For the distribution of the budget amount, the values entered corresponded to shares (in %). Although the distribution task is more complex (as opposed to a scale classification), it aligns more closely with the nature of the decision-making process.

To test H2 (“Change in levels of satisfaction due to COVID-19”), satisfaction with the individual types of healthcare services (F101; H2a) was determined on a scale from 0 (not satisfied at all) to 100 (perfectly satisfied) using a slider scale. Aspects that are important for increasing satisfaction (F102; H2b) were determined on a scale from 0 (absolutely unimportant) to 100 (extremely important) also using a slider scale. To test H3 (“Demand-oriented offers”), the distribution according to actual figures from the healthcare system was compared to the participants’ preferential distributions (F109).

## 3. Results

### 3.1. Willingness to Pay

The question here is about the available budget, regardless of how the available funds are allocated. Essentially, the healthcare budget results from the sum of health insurance contributions and is generally determined on the basis of actual cost trends. However, if contributors are willing to pay higher contributions, the budget to be distributed could also increase, regardless of the allocation.

To this end, participants were asked to state the maximum monthly amount that they would spend on all healthcare services (medical care, hospital care, therapy, medication, and rehabilitation measures) at an “optimum benefit level” (F104). Such expenditure might include health insurance contributions, as well as payments for optional services and supplementary benefits. The answers were, therefore, given in EUR and were divided into classes with a width of EUR 50. The results are presented in Figure 1.

The pandemic had an evident impact on the willingness to pay, as the mean value of payments increased from EUR 136.30/month to EUR 178.66/month (+31.1% vs. inflation +17.8%). The amount of WTP was very low in absolute terms but relatively high in relation to the income of the target group of students. As the distributions differed obviously, it can be concluded that the pandemic has had an impact on the willingness to pay. Consequently, higher contributions—and, thus, a higher budget—could be achieved as a basis for allocation.

### 3.2. Preferences and Their Changes Due to COVID-19 (H1)

**(H1a).** 
*The preferences for budget allocation to address diseases will change, and the COVID-19 pandemic has had an influence on such allocation.*


To obtain a demand- or preference-oriented distribution of healthcare budget per disease, the participants’ preferences were surveyed. A healthcare budget of EUR 100 billion was assumed, as this round figure is easy to process and allocate. Germany’s actual healthcare expenditure was EUR 415 billion in 2019 and EUR 498 billion in 2022 (+20%) [19]. The diseases surveyed corresponded to the 14 ICD groups. The points, therefore, had to be allocated to these 14 ICD groups, the sum of which was normalized ex-post to 100. Any *N*/*A* values in individual fields that were not filled in were replaced with 0.

The allocation changed only slightly for certain disease categories (infections, neoplasms, and “others”). For the majority of diseases (12 out of 14 categories), the overlapping 2-standard errors in Figure 2 suggest that the changes were not statistically significant. Even in the few categories without overlapping errors, the differences were relatively small. Overall, the ranking of disease categories remained largely consistent, and there are virtually no notable differences in the distribution between 2019 and 2023. The null hypothesis that the difference in diagnosis frequencies between 2019 and 2023 exceeds ±2.5 percentage points was rejected using the Two One-Sided Tests (TOST) procedure (*p* = 0.0198), indicating that the distributions can be considered statistically equivalent. This suggests that no practically relevant differences occurred in the distribution over time (In addition, the null hypothesis that there is no difference in diagnosis frequencies between 2019 and 2023 could not be rejected based on an independent samples *t*-test (*p* = 0.8836), suggesting that the distributions remained stable over time). Accordingly, Hypothesis H1a must be rejected, as preferences before and after the COVID-19 pandemic have remained largely unchanged.

**(H1b).** 
*The preferences for budget allocation to fund research into diseases will change, and the COVID-19 pandemic has had an influence on such allocation.*


As with the overall budget, allocation preferences for research budgets have not changed significantly. The overall rankings remained similar, with only minor differences in distribution between 2019 and 2023. In 14 out of 14 categories, the 2-sigma error margins overlapped, and the remaining categories showed similar values (Figure 3). The null hypothesis that the difference in diagnosis frequencies between 2019 and 2023 exceeds ±2.5 percentage points overall was rejected using the Two One-Sided Tests (TOST) procedure (*p* = 0.0405), indicating that the distributions can be considered statistically equivalent. This suggests that no practically relevant differences occurred in the distribution over time (In addition, the null hypothesis that there is no difference in diagnosis frequencies between 2019 and 2023 could not be rejected based on an independent samples *t*-test (*p* = 0.9345), suggesting that the distributions remained stable over time). Accordingly, Hypothesis H1b must be rejected, as the results for research budget preferences were even more consistent across the two periods than those for total budget allocations.

A direct comparison of the changes between 2019 and 2023 is shown in Figure 4.

The budget preferences changed slightly in terms of both the general budget and the research budget. However, the change was in the same direction in 10 out of 14 categories, indicating stable relative preference distributions.

### 3.3. Satisfaction (H2)

**(H2a).** 
*Satisfaction with the types of healthcare services has changed.*


In addition to the changes in preference in the context of the COVID-19 pandemic, the question regarding how general perceptions of the healthcare system may have changed also arises. This perception was measured using satisfaction levels, which were surveyed on a scale from 1 to 100. In this context, a significant change was interpreted as a deterioration. The objective is, therefore, to prove statistically that the COVID-19 pandemic did not result in a decline in satisfaction with health services. Participants rated these factors on a sliding scale from 1 (absolutely unimportant) to 100 (extremely important). Figure 5 summarizes the satisfaction of the participants with respect to different areas of the healthcare system.

Overall, the satisfaction levels were relatively high, with at least 60% approval reported across all healthcare services. Notably, satisfaction appears to have slightly increased following the COVID-19 pandemic. However, this increase was minimal, as indicated by overlapping 2-sigma error margins in 5 out of 6 categories, with medical technology being the only exception. The null hypothesis that there is no difference in service usage between 2019 and 2023 was rejected based on an independent samples *t*-test (*p* = 0.0295), indicating a statistically significant change over time (The null hypothesis that service satisfaction in 2023 is less than or equal to that in 2019 was rejected in a one-sided *t*-test (*p* = 0.0048), indicating a significant increase over time.). Therefore, Hypothesis H2a appears to be rejected, suggesting that no substantial changes in satisfaction have occurred. Medical technology may represent an exception, possibly due to rising expectations as technological advancements continue. Another focus of the study concerned the factors that may contribute to increased satisfaction with the healthcare system.

**(H2b).** 
*The relative importance of satisfaction measures has changed.*


The following measures could lead to the optimization of healthcare services. The higher the importance of the measure, the more sensitively patients react to this aspect. However, as sensitivity increases, so does the perception of the importance of healthcare quality.

All optimization factors were perceived as similarly important, with a particular emphasis on waiting times, quality of treatment, and additional insurance benefits (Figure 6). The importance of satisfaction-related measures appears to have increased slightly in nearly all cases following the COVID-19 pandemic. However, the 2-sigma error margins overlap across all categories, indicating that these differences are not very large. The null hypothesis that service satisfaction in 2023 is less than or equal to that in 2019 was rejected through the one-sided *t*-test (*p* = 0.0048), indicating a significant increase over time. As no changes in the sense of deterioration were observed, Hypothesis H2b must be rejected. The slight increase in the relative importance of satisfaction measures shows that the importance of the healthcare system is perceived as more sensitive and, therefore, more important.

### 3.4. Differences in Comparison to Actual Costs

#### 3.4.1. Change in Actual Costs

Next, the demand-oriented allocation preferences determined in this study were compared to the actual supply-oriented distributions of healthcare expenditure per illness. Expenditure data were available for approximately three years before the preferences were collected. The distribution of total healthcare expenditure, determined from Gesundheitsberichterstattung des Bundes (GBE, federal healthcare reporting) data, is shown in Figure 7.

A conditional change in the distribution is also evident here; for example, changes can be seen in the distributions for infections and neoplasms. The difference, in terms of expenditure in EUR million, is clear. As healthcare expenditure is factual and not the result of a survey, there are no fluctuations here. If we only consider the changes in expenditure, the picture shown in Figure 8 emerges.

Reductions are particularly evident for infections, eye care, and digestive system conditions, while increases can be observed for neoplasms and the treatment of psychological, nervous system, and “other” diseases.

#### 3.4.2. Comparison of Actual Costs vs. Preferences (H3)

**(H3).** 
*Demand preferences and actual healthcare expenditure are in line (i.e., equal).*


To identify the concrete differences between the allocation preferences (P) and the actual expenditure (E), the corresponding pre- and post-COVID-19 differences (i.e., E2015–P2019 and E2020–P2023) were compared (Figure 9). Here, negative values indicate that the proportional preferences are higher than the actual expenditure and vice versa.

The graph clearly shows that preferences were often rated higher than expenditure, particularly in the areas of infections, neoplasms, endocrine treatments, and ear and skin conditions, resulting in negative differences. In contrast, for mental illness, circulatory, digestive, musculoskeletal, and especially “other” conditions, positive differences emerged, indicating higher expenditure than preferences. Both the one-sample *t*-test and the Wilcoxon signed-rank test indicated that the absolute differences between preferences and budgets significantly exceeded the predefined threshold of 1 percentage point in both 2019 and 2023 (*t*_(2019)_ = 5.32, *p* < 0.0001; *t*_(2023)_ = 5.56, *p* < 0.0001; W_(2019)_ = 89.0, *p* = 0.0004; W_(2023)_ = 90.0, *p* = 0.0002), suggesting consistent and robust discrepancies between demand-oriented preferences and supply-oriented actual expenditures.

The particular differences in the unspecified “other” diseases can be broken down in terms of content as follows:D50–D90: Diseases of the blood and blood-forming organs and certain disorders involving the immune system;O00–O99: Pregnancy, childbirth, and puerperium;P00–P96: Certain conditions originating in the perinatal period;Q00–Q99: Congenital malformations, deformities, and chromosomal abnormalities;R00–R99: Symptoms and abnormal clinical and laboratory findings not elsewhere classified;N00–N99: Diseases of the genitourinary system;Z00–Z99: Factors influencing health status and leading to health care use.

However, it is also obvious that the differences before and after the pandemic are in the same direction for all categories (i.e., uniformly positive or negative in each case). The differences between preferences and current spending are, therefore, clearly stable. This also means that changes in spending would make sense if an approximation of the preferences were implemented.

A question then arises: what effect has the COVID-19 pandemic had on changes in preferences and health expenditures in each disease category? A comparison of the directions of change, in terms of actual costs and preferences, is provided in Table 2.

In as many as 9 out of 14 cases/diseases (i.e., in almost half of the cases), the preferences developed in the opposite direction to the change in actual expenditure. This means that COVID-19 has further increased some of the existing differences between preference and expenditure while reducing others.

With regard to these different directions (and resulting increased differences), a further distinction can be made between cases with increased expenditure and lower expectations/preferences (i.e., cancer, mental illnesses, and nervous systems) and those with reduced expenditure and higher expectations (e.g., infections, eye care, and skin conditions).

Overall, due to the differences between preferences and expenditure, Hypothesis H3 must be rejected, and the differences considered must be confirmed.

### 3.5. Average Costs

In addition to total expenditure in the healthcare system and its allocation to individual diseases, the cost of treating individual cases is also relevant to the distribution of expenditure. This individual expenditure is not directly or officially reported; however, it can be approximated if the number of cases is known. However, a distinction must be made between outpatient and inpatient cases. As outpatient and inpatient treatment are structurally different, there are also fundamental differences, meaning that average costs cannot be calculated directly. This is particularly evident in the other forecasts. Nevertheless, clear differences between budget and case distributions can indicate conclusions regarding special effects (see also Figure 10).

In some cases, the expenditure share is much higher than the share of cases. This is particularly true for psychological conditions and, to a lesser extent, for digestive and nervous system treatments. This discrepancy is extreme for psychological illnesses and “other” diseases, which can possibly be explained by the fact that treatment for psychological illnesses is usually relatively long-term, therefore resulting in relatively high expenditures. In some cases, there are also clear differences between outpatient and inpatient treatment, which may be due to the type of illness. For example, cancer and circulatory problems such as heart attacks are typically treated on an inpatient basis. Consequently, further analyses should also consider the differences between outpatient and inpatient treatment in more detail.

## 4. Discussion

In addition to the analysis of the results, two important key points need to be discussed. This study showed only a minor COVID-19-induced impact on demand in Germany’s healthcare sector. Due to the significance of the COVID-19 pandemic, a strong preferential shift toward funding for COVID-19-related illnesses would not have been surprising; however, this was not observed, and the demand picture remained relatively stable.

A match between the services demanded, and the services offered would occur in the presence of exclusive market mechanisms and/or in an optimally regulated system that takes patient preferences into account. However, such a match was not observed here, and there were clear deviations. Further analysis would certainly be necessary to investigate this question further, which could help to achieve a more patient demand-oriented allocation through regulatory policy. The discrepancy between actual demand and supply identified here needs to be resolved.

As mentioned above, further detailed information (e.g., treatment costs and fairness considerations) can play a role in the allocation of resources and budgets [23]. Other preference-influencing criteria could include “severity of illness, age, daily care needs, number of alternative interventions, [the] economic status of the person and illnesses with work absences” [24]. Diederich et al. have cited the life-threatening and acute nature of illnesses as essential information affecting the prioritization of individual cases [3]. It is also notable that preferences in individual cases can depend significantly on a patient’s age, gender, and income [25]. Furthermore, moral aspects (e.g., the prioritization of a more youthful, healthier lifestyle [4]), quality-of-life considerations, and the introduction of new technologies may all also have effects on allocation [26]. Aspects of bounded rationality (e.g., equality aspects, the repetition of decisions) can also play a certain role in this context [27]. In this respect, distortions may arise in the aggregated results. As complete information on other free markets, such as the stock market, is also unavailable, restriction to the aggregate seems sufficient. However, the detailed aspects listed above offer a means for optimization of such analyses.

As a limitation, it should be noted that this experimental preliminary study was based on a limited sample of potential future contributors and, therefore, is not representative of the general population. In the specific experimental group surveyed, the focus was not explicitly on how the respondents were particularly affected by COVID-19 but rather on the preferences of young future contributors. The preliminary study clearly showed that clear results are discernible for the considered sample and, therefore, a more comprehensive main study appears to make sense. Such a planned main study is expected to provide more strongly differentiated results than those detailed here. Nevertheless, this study remains relevant, as it provides insights into the preferences of a target group that holds considerable significance.

A further limitation is that the actual expenditure budget data were, in each case, around 3–4 years before the questionnaire was issued, so it was assumed that expenditure had hardly changed at this time. This relative stability was assumed as the expenditure distributions are only measured every 5 years and, therefore, no annual data are available, given that the government obviously sees no need for annual measurements. Furthermore, despite it being obtained at the peak of the pandemic, the 2020 expenditure did not yet reflect all the associated effects. Therefore, the main study should be carried out after the next expenditure survey, allowing the results to be updated with respect to the new figures. Nevertheless, it would generally be better to synchronize the expenditure and survey more closely.

In addition, the results obtained in the German context can be compared to international results, taking into account the needs and practices of other countries’ healthcare systems. The operationalization of the inclusion of patient preferences in the allocation of healthcare budgets can be achieved, for example, by considering key decisions such as the distribution of healthcare infrastructure or the definition of the catalog of services by the FJC.

In summary, this study’s limitations are related to its chosen design, as a questionnaire study with a limited number of participants does not necessarily provide a representative result. Therefore, we plan to conduct more comprehensive studies in the future.

## 5. Conclusions

The assessment of demand-side preferences revealed, on the one hand, that these preferences remained relatively stable, even in the face of major events such as the COVID-19 pandemic. This suggests that incorporating public preferences into allocation decisions would not result in continuous shifts or significant administrative burdens. On the other hand, the findings highlighted clear discrepancies between actual healthcare expenditure and the preferences of insured individuals, especially in areas such as cancer, infections, and mental health. This underscores the potential value of preference measurement as a key instrument for guiding resource allocation and avoiding misallocations. In this context, misallocation refers to a distribution of funds in which the benefit to the insured population could be increased through reallocation—an objective that should be important in the context of any healthcare system. As it was not the aim of our study to determine the demand exactly but, instead, to show that it is important to do so, we believe that this paper provides evidence in this line. Future research in the healthcare field should dedicate sufficient effort to determine the demand, which can be expected to improve the supply side through optimized budget allocation.

The COVID-19 pandemic has not eliminated misallocations. The impacts of the COVID-19 pandemic on both changes in preference and actual allocation are based on expected disease patterns but were found to be weaker than expected. As stated above, preferences appeared to remain relatively stable.

Similarly, satisfaction with the healthcare system did not decline in the wake of the pandemic; in some areas, it even increased slightly. This may be attributable to stability in both patient preferences and actual expenditure. Alternatively, it may be due to the enhanced perception of the state as one that cares for its citizens. Finally, the presented findings underscore a notable willingness among individuals to contribute financially to the funding of healthcare services.

Despite the limitations inherent to a single study with a limited number of participants, the findings clearly suggest that behavioral factors—such as preferences and satisfaction—should play a more prominent role in healthcare budget allocation. In this regard, the collection of more detailed and representative data would be valuable for further validation of the results of this initial experimental study.

## Figures and Tables

**Figure 1 healthcare-13-01074-f001:**
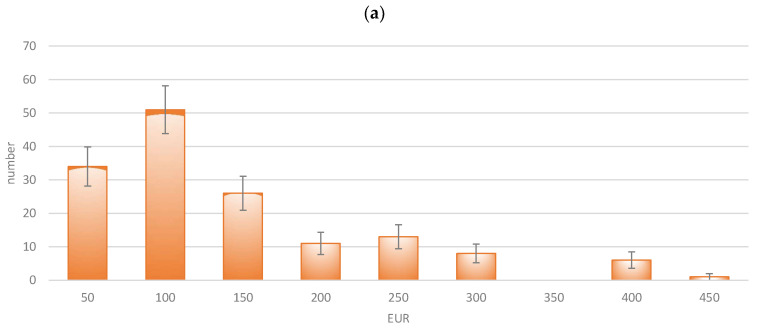
Willingness to pay for health expenditure (maximum monthly amount in EUR): (**a**) 2019; (**b**) 2023.

**Figure 2 healthcare-13-01074-f002:**
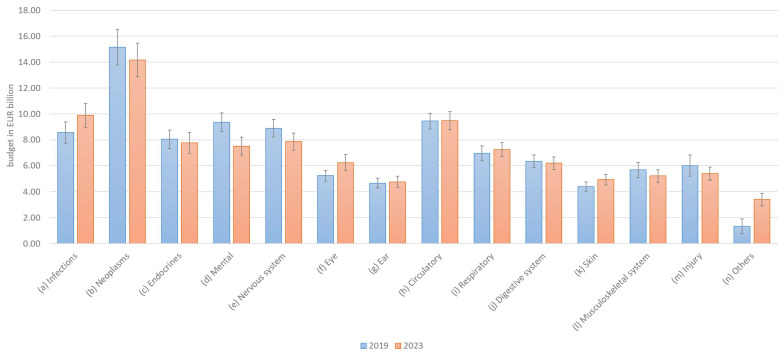
Preferences for allocation of health budgets by disease category.

**Figure 3 healthcare-13-01074-f003:**
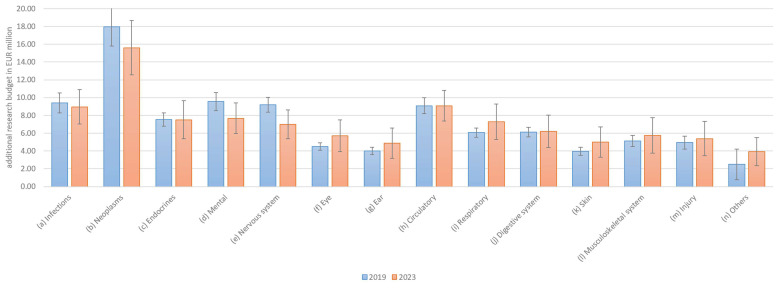
Preferences for allocation of research budgets by disease category.

**Figure 4 healthcare-13-01074-f004:**
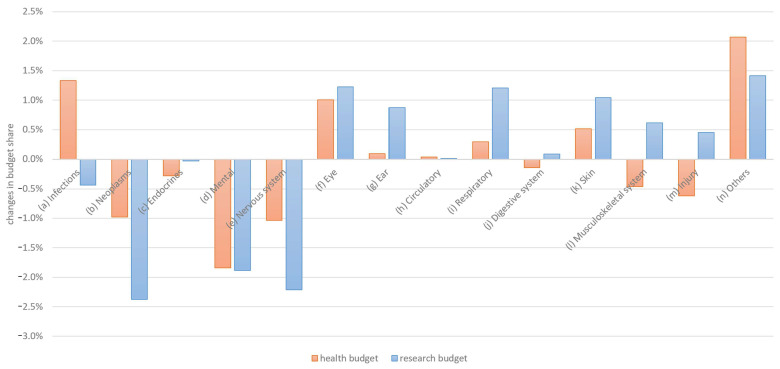
Changes in general and research budget allocation preferences by disease category.

**Figure 5 healthcare-13-01074-f005:**
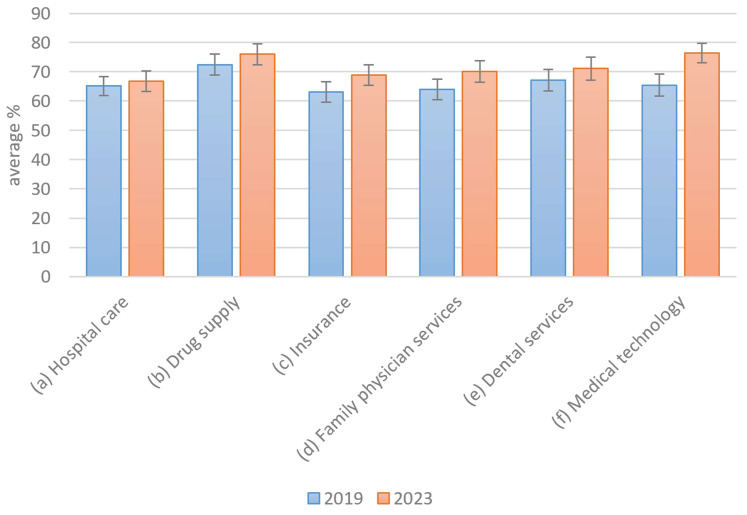
Satisfaction with healthcare services.

**Figure 6 healthcare-13-01074-f006:**
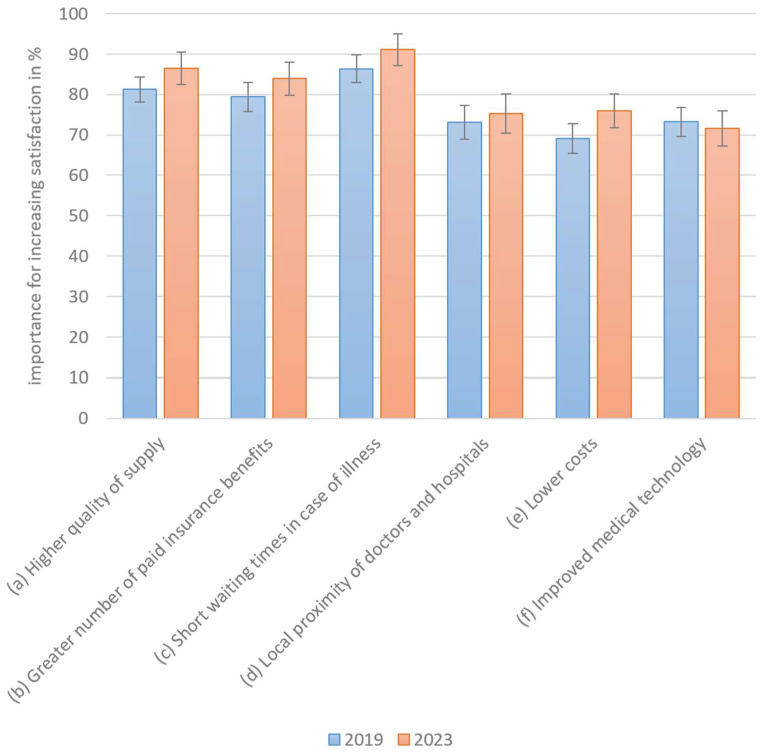
Measures of importance for increasing satisfaction.

**Figure 7 healthcare-13-01074-f007:**
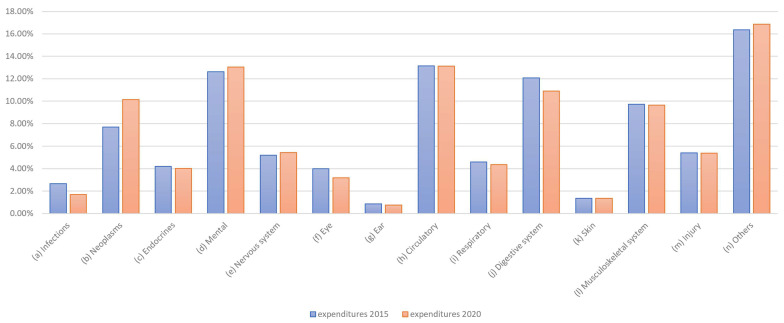
Share of health expenditure by disease category.

**Figure 8 healthcare-13-01074-f008:**
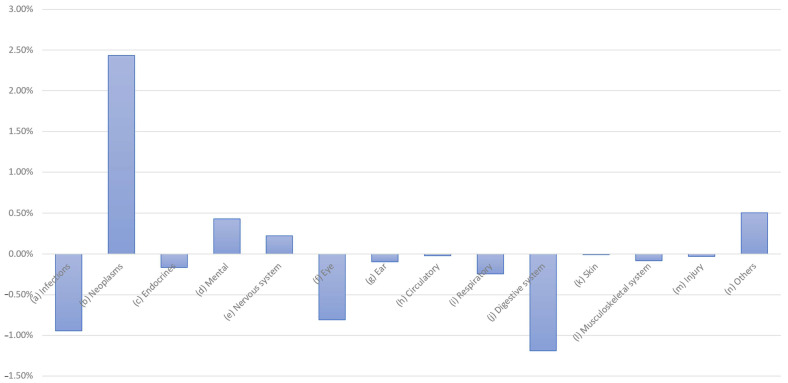
Changes in health expenditure by disease category, 2015 vs. 2020.

**Figure 9 healthcare-13-01074-f009:**
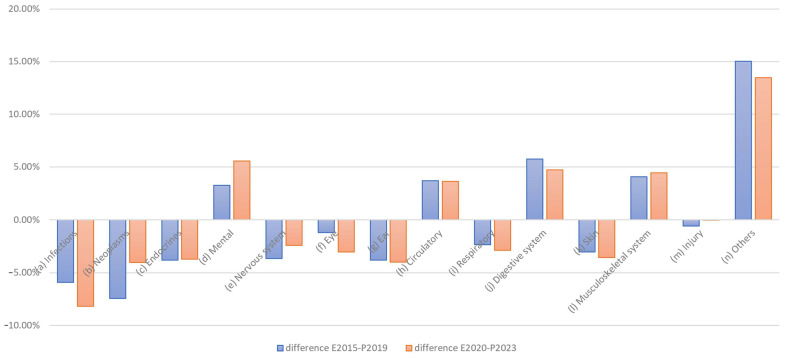
Differences between expenditures and preferences.

**Figure 10 healthcare-13-01074-f010:**
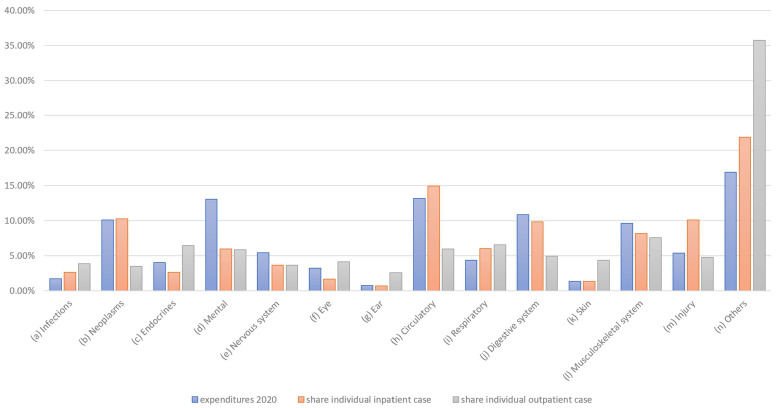
Share of expenditure per case and inpatient/outpatient case numbers.

**Table 1 healthcare-13-01074-t001:** Participant characteristics.

Characteristic		2019		2023	
Participants	n	157		163	
Gender	Male n [%]	91	[58.0%]	87	[53.4%]
	Female n [%]	65	[41.4%]	75	[46.0%]
	Others n [%]	1	[0.6%]	1	[0.6%]
Age	Mean [SD]	28.268	[11.83]	31.654	[14.94]
	19–29 years	73.25%		59.26%	
Household income per year (€ thousand)	Mean [SD]	20.49	[22.39]	34.14	[38.67]
Total hospital stays	Mean [SD]	3.10	[3.48]	3.93	[4.60]
Doctor visits per year	Mean [SD]	3.75	[4.11]	4.93	[4.75]

**Table 2 healthcare-13-01074-t002:** Comparison of changes in preferences and health expenditures.

	Preferences			Health Expenditures			Changes		
	2019	2023	Changes	2015	2020	Changes	Preferences	Expenditures	Same Direction
(a) Infections	8.55%	9.89%	1.34%	2.66%	1.71%	−0.94%	+	−	no
(b) Neoplasms	15.15%	14.17%	−0.98%	7.71%	10.14%	2.44%	−	+	no
(c) Endocrines	8.04%	7.76%	−0.28%	4.20%	4.04%	−0.17%	−	−	yes
(d) Mental	9.35%	7.51%	−1.84%	12.63%	13.06%	0.43%	−	+	no
(e) Nervous system	8.89%	7.86%	−1.03%	5.21%	5.44%	0.22%	−	+	no
(f) Eye	5.23%	6.23%	1.01%	4.01%	3.20%	−0.81%	+	−	no
(g) Ear	4.66%	4.76%	0.09%	0.86%	0.76%	−0.10%	+	−	no
(h) Circulatory	9.44%	9.47%	0.03%	13.16%	13.14%	−0.02%	+	−	no
(i) Respiratory	6.95%	7.24%	0.30%	4.60%	4.36%	−0.25%	+	−	no
(j) Digestive system	6.34%	6.20%	−0.14%	12.09%	10.90%	−1.19%	−	−	yes
(k) Skin	4.41%	4.92%	0.51%	1.37%	1.36%	−0.01%	+	−	no
(l) Musculoskeletal system	5.66%	5.20%	−0.46%	9.74%	9.65%	−0.08%	−	−	yes
(m) Injury	6.00%	5.38%	−0.62%	5.41%	5.38%	−0.03%	−	−	yes
(n) Others	1.34%	3.40%	2.07%	16.36%	16.86%	0.51%	+	+	yes

Note: + = increase, − = decrease.

## Data Availability

The used dataset is available at https://osf.io/b79wf (accessed on 5 May 2025).

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
