# Peer review of "How Stable Are Individual Preferences for Health Expenditure in Germany?"

_healthcare, 2025, doi:10.3390/healthcare13091074_

Round 1
Reviewer 1 Report
Comments and Suggestions for Authors
This manuscript addresses a highly relevant and important topic: the alignment between public preferences for healthcare resource allocation and actual expenditure patterns in Germany. The inclusion of the COVID-19 pandemic as a potential disrupting event adds a timely and interesting dimension. The study employs a longitudinal design comparing pre- and post-pandemic preferences and satisfaction, and compares these preferences to official expenditure data. While the research question is pertinent and the basic design has merit, the study suffers from significant methodological limitations, primarily concerning sample representativeness and the temporal alignment/comparability of preference and expenditure data, which severely impact the generalizability and robustness of the study. The finding of stable preferences despite the pandemic is notable, as is the persistent gap between preferences and actual spending.
Major weaknesses
- The participants were recruited via university seminars and consisted largely of students or young adults (mean age ~28-32) with higher-than-average education levels. This sample is notrepresentative of the general German population, taxpayers, or insured individuals. While the authors justify this by focusing on "future payers," this does not allow for valid generalizations about the currentpreferences or satisfaction of the overall population, which is implied throughout the introduction and discussion. Conclusions drawn about alignment with national healthcare expenditure are therefore highly questionable.
- Preferences were measured in April 2019 and April 2023. Actual expenditure data used for comparison are from 2015 and 2020. Comparing 2019 preferences with 2015 expenditure involves a 4-year gap. Comparing 2023 preferences with 2020 expenditure involves a 3-year gap. The assumption that expenditure proportions remained largely unchanged between 2015-2019 and 2020-2023 needs much stronger justification or needs to be acknowledged as a major limitation potentially confounding the comparison (H3). While 2020 captures earlypandemic costs, it might not fully reflect the longer-term shifts potentially influencing 2023 preferences.
- The rejection of hypotheses H1a, H1b, H2a, and H2b appears to be based primarily on visual inspection of graphs and the overlap of standard error bars (or implied 2-sigma margins). While this provides an initial indication, formal statistical tests (e.g., t-tests, ANOVA, MANOVA, Chi-squared tests as appropriate for distributional comparisons) are required to rigorously test for statistically significant differences between the 2019 and 2023 groups for each preference/satisfaction measure. Relying solely on error bar overlap (especially standard error) is insufficient for hypothesis rejection.
- Section 3.5 compares expenditure shares with case number shares, but crucially relies only on inpatient case numbers from 2021. This is a flawed approach as it ignores the vast number of outpatient cases, which vary significantly across disease categories. This will heavily distort the calculated "cost per case share," likely inflating it for inpatient-heavy conditions and deflating it for outpatient-heavy ones. The conclusions drawn in this section are unreliable due to this data limitation. The assumption that inpatient numbers are a stable proxy across all disease categories for total cases/costs is unlikely to hold.
- Due to the non-representative sample, the findings cannot be reliably generalized to the German population. The study provides insights into the preferences and satisfaction of a specific young, educated cohort, but claiming these reflect broader public sentiment or provide a firm basis for national policy changes is unwarranted.
Minor weaknesses
- Section 2.3 describes F102 as measuring satisfaction using a scale, while Section 3.3 (H2b) refers to the "Importance of satisfaction measures" and implies higher scores mean more dissatisfaction or demand for optimization. This is confusing. Please clarify precisely what was asked (Importance? Need for improvement? Satisfaction with the potential for optimization?) and ensure the interpretation in the results aligns consistently. Figure 6 is labeled "Importance," but the text suggests it measures demand/dissatisfaction.
- Phrases like "appears to be rejected" (used frequently when discussing hypotheses) should be replaced with conclusions based on statistical testing (e.g., "no statistically significant difference was found," leading to failure to reject the null hypothesis, or "a significant difference was observed").
- While limitations are mentioned, the impact of the non-representative sample needs to be more strongly emphasized and integrated throughout the discussion and conclusions, rather than primarily mentioned at the end. The implications of the temporal data mismatch also need fuller discussion.
Author Response
Dear Reviewer,
Thank you very much for your thoughtful and constructive comments. They were extremely helpful in substantially improving our manuscript. We fully agree with all of your suggestions and have implemented them through appropriate additions, corrections, and more detailed explanations and clarifications.
Furthermore, the manuscript has been revised by a professional English Language Editing Service. You will find our detailed point-by-point responses to your comments in the attached Word document.
In addition, we have slightly clarified individual explanations and resolved minor technical issues. We hope that our revisions adequately reflect your comments and have helped to further improve the quality of the manuscript. Once again, thank you very much for your valuable support!
Best regards,
Bodo Vogt & Andreas Uphaus

Reviewer 2 Report
Comments and Suggestions for Authors
Linguistic Corrections and Suggestions
Key issue
Suggestion: Key Issues
(The plural form is more appropriate given the range of topics addressed.)
Lines 135–137, 261, and 467 could benefit from improved linguistic fluency.
Expressions such as “healthcare system” and “preferences” appear repeatedly and in close proximity throughout the text. A stylistic review could help improve the overall flow and avoid redundancy.
Consider standardizing terminology: for example, the text alternates between “the Covid pandemic” and “COVID-19 pandemic.” Choose one and apply it consistently throughout the manuscript.
Abstract
The Methods section should be improved. More detail is needed regarding the questionnaire, including its structure and how it was administered.
The Conclusion also needs to be strengthened to better reflect the study’s main findings and implications.
Main Manuscript
The theoretical framework should be expanded. Issues such as the funding of the healthcare system must be clearly explained from the outset. The German healthcare model, still largely based on the Bismarck system, should be introduced and described, especially for international readers unfamiliar with it.
- For example, in line 218/219, funding is mentioned without prior clarification, which may confuse readers.
The importance and influence of patient-centred care models and the inclusion of patient preferences are not clearly addressed and should be elaborated upon.
The methodology section requires a thorough revision:
- The questionnaire is not described in sufficient detail, nor is there any mention of its validation.
- The formulation of hypotheses does not follow standard academic conventions.
- The sampling strategy must be clearly explained and detailed.
- The respondents, who are predominantly young students, were selected without a clear description of inclusion criteria.
- This raises concerns about sampling bias, especially given that the elderly were among the most affected groups during the pandemic.
- It is therefore essential to justify and explain the sample selection criteria.
Conclusions
The conclusions should highlight that there are notable discrepancies between individual preferences and actual healthcare spending, especially in areas such as infectious diseases, cancer, and mental health.
They should also state that overall satisfaction with the healthcare system did not decline during the pandemic—in some areas, it even increased slightly.
Finally, it should be emphasized that individuals are willing to contribute financially to support healthcare services.
Author Response

(The authors gave the same response as above.)

Round 2
Reviewer 1 Report
Comments and Suggestions for Authors
Thank you for your contribution to address my comments for improving my comments.
Reviewer 2 Report
Comments and Suggestions for Authors
I look forward to the continuation of this study with a different type of sampling. The recommendations made for this article should be followed.